# Travel Time Reliability-Based Rescue Resource Scheduling for Accidents Concerning Transport of Dangerous Goods by Rail

Lanfen Liu and Xinfeng Yang *

School of Traffic and Transportation Engineering, Lanzhou Jiaotong University, Lanzhou 730070, China; liulanfen@mail.lzjtu.cn
* Correspondence: xinfengyang@mail.lzjtu.cn

**Abstract:** The characteristics of railway dangerous goods accidents are very complex. The rescue of railway dangerous goods accidents should consider the timeliness of rescue, the uncertainty of traffic environment and the diversity of rescue resources. Thus, the purpose of this paper is to confront the rescue resources scheduling problem of railway dangerous goods accident by considering factors such as rescue capacity, rescue demand and response time. Based on the analysis of travel time and reliability for rescue route, a multi-objective scheduling model of rescue resources based on travel time reliability is constructed in order to minimize the total arrival time of rescue resources and to maximize total reliability. The proposed model is more reliable than the traditional model due to the consideration of travel time reliability of rescue routes. Moreover, a two-stage algorithm is designed to solve this problem. A multi-path algorithm with bound constraints is used to obtain the set of feasible rescue routes in the first stage, and the NSGA-II algorithm is used to determine the scheduling of rescue resources for each rescue center. Finally, the two-stage algorithm is tested on a regional road network, and the results show that the designed two-stage algorithm is valid for solving the rescue resource scheduling problem of dangerous goods accidents and is able to obtain the rescue resource scheduling scheme in a short period of time.

**Keywords:** rescue resource; multi-resource scheduling; NSGA-II algorithm; reliability; multi-objective; railway dangerous goods accident



## 1. Introduction

Dangerous goods include the substances and articles the carriage of which is prohibited by the Regulation Concerning the International Carriage of Dangerous Goods by Rail (RID) or authorized only under the conditions prescribed therein [1]. Specific regulations are in place in relation to the transport and storage of dangerous goods, as well as employee, consumer and environmental protection, in order to prevent accidents during carriage of such goods.

Transport of dangerous goods is regulated in order to prevent, as much as possible, accidents related to persons or property and damage to the environment, the means of transport employed or to other goods. The transportation of dangerous goods (TDG) includes activities related to the movement of dangerous goods from their place of manufacture or storage to their destination with the preparation of cargo, packaging, vehicles and crew, reception of goods, carrying out cargo operations and short-term storage of goods at all stages of their transfer [2].

Currently, a large quantity of different types of dangerous goods is transported by road, rail, inland waterways and maritime, pipelines and air. Among them, railway transportation undertakes the main traffic volume. The influence of random factors and events can result in an accident resulting in leakage, combustion or explosion of hazardous substances during loading and unloading, transportation and storage procedures. These types of incidents not only threaten the safety of rail transport but also the life, environment

and property [3]. The most common places where dangerous goods accidents happen are dangerous goods handling locations of petrochemical and logistics enterprises, railway stations and lines along the railway. Moreover, these locations have many internal equipment and facilities, and their external rescue environment is very complex. Therefore, once an accident occurs, it is necessary to carry out emergency rescue in a timely manner; otherwise, it will cause serious consequences.

There has been a substantial amount of investigations on emergency resource scheduling or its related problems. Emergency resource scheduling is the scheduling optimization operation spanning across the categories of routing and emergency materials distribution.

In emergency relief research, reviews by Caunhye et al. [4], Galindo and Batta [5] and Zhou et al. [6] revealed that most of the models developed for emergency relief only permit scheduling of one type of resource, which is either expendable or non-expendable [7]. Özdamar et al. [8], Chang et al. [9], Balcik et al. [10] and Huang et al. [11] only considered resource scheduling related to expendable resources. In contrast, Lassiter et al. [12], Wex et al. [13] and Schryen et al. [14] only considered resource scheduling related to non-expendable resources in disaster emergencies.

Some researchers have also proposed models for multi-resource scheduling of emergency relief. Zheng and Gao [15] built a multi-resource emergency systems model with consumption rates of nonnegative and integrable functions based on the object of earliest emergency-start-time, and the corresponding algorithm was provided. Lee et al. [16] studied a scheduling problem with operations that required renewable as well as non-renewable resources and proposed a framework of heuristic procedures for solving this problem. Zhang et al. [17] built an emergency resource scheduling model, which included multiple suppliers with a variety of resources, a single accident site and some restrictions, and an adaptively mutate genetic algorithm is applied to figure out a superior solution. Shahparvari and Bodaghi [18] developed a mixed integer programming model to support tactical decision making in allocating emergency relief resources in the context of the Black Saturday bushfires. Furthermore, a possibilistic programming approach was employed to minimize the transportation disruption risk. In addition, Bodaghi et al. [19] analyzed several probabilistic scenarios to determine the most frequent emergency operation plan and the most persistent best compromised emergency operation plan. Li [20] constructed a resource scheduling model by considering influencing factors such as transportation time, cost and resource availability. Particle swarm optimization combined with Cuckoo search (PSO-CS) algorithm was applied to solve a resource scheduling process of multiple types and multiple classes. Bodaghi et al. [7] formulated the Multi-Resource Scheduling and Routing Problem (MRSRP) for emergency relief and developed a solution framework to effectively deliver expendable and non-expendable resources in emergency recovery operations. Tang and Sun [21] established two optimal models, respectively; the single objective model aimed at minimizing the time of emergency resource dispatch, and the multi objective model was for the sake of minimizing the time of emergency resource dispatch and the number of emergency rescue bases, and a voting analytic hierarchy process was used to solve the model. Wu and Xiang [22] constructed a capacitated vehicle routing problem model suitable for emergency resource scheduling process and designed a simulated annealing-genetic algorithm (SA-GA).

Many of the models developed for emergency resources scheduling are deterministic and had the minimization of travel time as one of the objectives; they ignored the rescue routing problem. Moreover, heuristic methods are always used to find a near-optimal due to the NP-hard nature of the problem. For example, the PSO-CS algorithm was applied to solve the resource scheduling process of multiple types and multiple classes by Li [20]. An adaptively mutate genetic algorithm (GA) was used to solve an emergency resource scheduling model and included multiple suppliers with a variety of resources by Zhang et al. [17]. SA-GA was proposed in order to confront the capacitated vehicle routing problem for emergency resource scheduling process by Wu and Xiang [22]. Wu et al. [23] used GA to find out the dynamic emergency distribution path. Zidi et al. [24]

and Mguis et al. [25] proposed a multi-agents approach by using a genetic algorithm for scheduling vehicle routing and local search for the management of an eventual event. Moreover, Liu et al. [26] designed a quick heuristic algorithm in order to obtain a fleet dispatching plan. Ceselli et al. [27] proposed an algorithm based on the branch-and-cut-and-price paradigm in order to deal with the generalized location and distribution problem. Tang et al. [28] used the greedy algorithm to optimize the distribution path. Zhao et al. [29] proposed a two-stage shortest path algorithm, which is composed of K-paths algorithm and shuffled frog leaping algorithm, in order to solve the dynamic paths planning problem of emergency vehicles. Özdamar and Demir [30] designed a multi-level clustering algorithm for coordinating vehicle routing in large-scale post-disaster distribution and evacuation activities. He et al. [31] adopted the K-means clustering algorithm in order to obtain some of the local distribution centers and used the PSO algorithm to design the local optimal allocation routings of emergency relief vehicles. Vargas-Florez et al. [32] used the hierarchical ascending clustering approach in order to discover the reliable delivery routes. Gharib et al. [33] proposed the NSGA-II and multi-objective firefly algorithm in order to deal with the multiobjective vehicle routing model that was developed. Penna et al. [34] proposed a generic hybrid heuristic framework to solve the vehicle routing problem after a natural disaster.

All these related investigations have given us much inspiration when studying the rescue resource scheduling problem. For dangerous goods accidents of railways, the rescue resource scheduling should meet the following requirements:

1.  The rescue resource scheduling problem is one of time urgency, and the rescue resources should arrive at the accident location in time. Since time is critical in railway dangerous goods accidents and short response time can help save more properties and lives and reduce the impact of accidents, the rescue work should be started at the first time of incidence. Therefore, the earlier the rescue resources are transported to the accident location, the better the rescue effect.
2.  Railway dangerous goods accident has continuous expansibility, and emergency rescue has time window requirements. Expansion of accident is the main feature of railway dangerous goods accidents, and it very easily affects other equipment or facilities and causes accident expansion. The characteristics of accident expansion require that rescue resources must be transported to the accident location within the time window before accident expansion.
3.  The traffic environment is uncertain, and rescue time reliability is an important quality performance measure of the rescue resource scheduling problem. Road traffic conditions are more complex and uncertain, which requires that the arrival time of rescue vehicles must be reliable.
4.  Railway dangerous goods accidents require coordinated dispatching of multi-rescue resources. The characteristics of railway dangerous goods determine the multiplicity of accident consequences. Therefore, accident rescues of railway dangerous goods generally require multi-rescue resources of fire control, medical and health.

Therefore, in the case of a railway dangerous goods accident, transportation of rescue equipment (such as fire truck, fire sprinkler, fire water monitor, emergency lighting equipment, chemical rescue vehicle, special rescue vehicle, medical rescue vehicle, etc.) and resources (such as rescue medicine, air respirator, etc.) to the accident location is urgently required. At the same time, due to the influence of rescue capacity, location, road network, traffic environment and traffic flow of each rescue center, the appropriate rescue resource scheduling and routing plans need to be integrated and reasonably arranged.

The remainder of the paper is organized as follows: Some mathematical notations are defined in Section 2.1 and the travel time and reliability of a rescue route are analyzed in Section 2.2. The constraints and objective function are discussed, and then a multi-objective scheduling model of rescue resources is established in Section 2.3. A multi-path algorithm with a bound constraint as the first stage of the two-stage algorithm is proposed in Section 3.1, and the NSGA-II algorithm as the second step is designed in Section 3.2.

The proposed algorithm is tested on a regional road network in Section 4. Finally, the conclusions are provided in Section 5.

## 2. The Reliability Based Rescue Resource Scheduling Model

In this section, a multi-objective optimization model of rescue resources is developed by considering the travel time and reliability of the rescue route. The travel time of links and intersection delay, the travel time and reliability of the rescue route are analyzed based on the mathematical notations defined in Section 2.1. Moreover, the constraints of resource demand, rescue capacity and flow balance, etc., are analyzed, and two objective functions are formulated in Section 2.3. The flow chart of this section is presented in Figure 1.

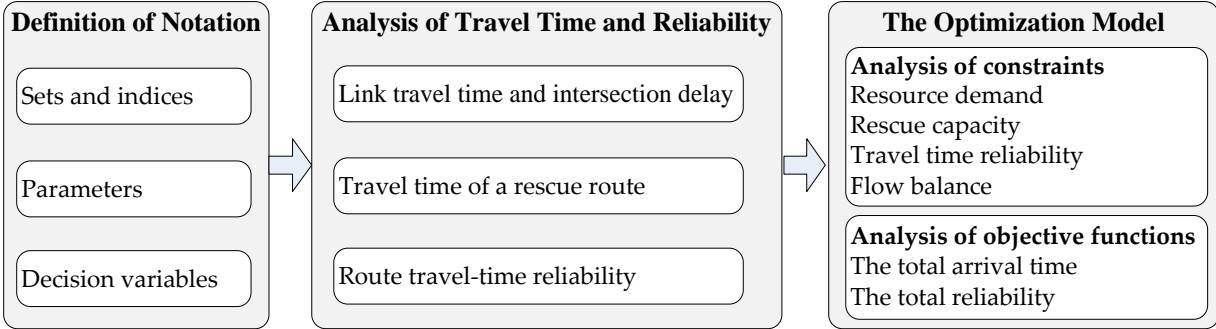

**Figure 1.** The flow chart of Section 2.

### 2.1. Notations

Variables and symbols in the model are defined as follows:

### 2.1.1. Sets and Indices

The sets and indices in this paper are listed in Table 1.

**Table 1.** Sets and indices in the model.

| Sets/Indices | Description |
| --- | --- |
| $I$ | Set of all resource types |
| $i$ | Index of rescue resource types, $i \in I$ |
| $R$ | Set of rescue centers |
| $r$ | Index of rescue centers, $j \in J$ |
| $s$ | The accident location |
| $E$ | Set of links |
| $e_{mn}$ | Index of links, $e_{mn} \in E$, $m, m \in V, m \neq n$ |
| $V$ | Set of intersections |
| $v$ | Index of intersections, $v \in V$ |
| $K_{rs}$ | Set of rescue routes between rescue center $r$ and accident location $s$ |

### 2.1.2. Parameters

The parameters of rescue demand and rescue center are listed in Table 2.

**Table 2.** Parameters of rescue demand and rescue center.

| Parameters | Description |
| --- | --- |
| $ED_i$ | The demand for a rescue resource type $i$ |
| $RT_i$ | Maximum tolerable transport time for a resource type $i$ |
| $EN_{ir}$ | Capacity of rescue center $r$ for a resource type $i$ |
| $t1_{ir}$ | Assembly time of rescue resource $i$ of rescue center $r$ |
| $t_{mn}$ | Travel time of link $e_{mn}$ |
| $d_v$ | Delay of intersection $v$ |

### 2.1.3. Decision Variables

The decision variables are listed in Table 3.

**Table 3.** Decision variables.

| Parameters | Description |
|---|---|
| $EM_{ir}$ | The supply quantity of rescue center $r$ for resource type $i$ |
| $k_{rs}$ | Index of routes between rescue center $r$ and accident location $s$ |
| $t2_{k_{rs}}$ | Travel time of route $k_{rs}$ from the rescue center $r$ to the accident location s |
| $R_{k_{rs}}$ | Travel time reliability of route $k_{rs}$ from the rescue center $r$ to the accident location s |
| $\delta_{mn}^{k_{rs}}$ | A binary variable. $\delta_{mn}^{k_{rs}} = 1$ means link $e_{mn}$ on route $k_{rs}$, otherwise $\delta_{mn}^{k_{rs}} = 0$ |
| $\delta_{v}^{k_{rs}}$ | A binary variable. $\delta_{v}^{k_{rs}} = 1$ means intersection $v$ on route $k_{rs}$, otherwise $\delta_{v}^{k_{rs}} = 0$ |

### 2.2. Analysis of Travel Time and Reliability

Due to the influence of various random events, such as weather conditions, traffic jams and traffic incidents, the supply of a traffic network may be uncertain, which varies the travel time of a rescue route randomly. The different positions of different rescue centers result in the differences in travel time and reliability of rescue routes. Therefore, it is necessary to analyze the travel time of a rescue route in rescue resource scheduling.

#### 2.2.1. Link Travel Time and Intersection Delay

For a road section, its travel time is mainly affected by link traffic flow and intersection delay. The travel time $t_{mn}$ of rescue vehicles on a link mainly includes two parts: the travel time of vehicles passing through link $e_{mn}$ under free-flow conditions, $t_{mn}^0$, and the delay time on the link $e_{mn}$ due to vehicle speed fluctuation, $\xi_{mn}$. Thus, the travel time of a link can be expressed as the following:

$$t_{mn} = t_{mn}^0 + \xi_{mn} \tag{1}$$

where delay time $\xi_{mn}$ is a random variable that is related to the average speed of vehicles, vehicle density and time headway.

Rescue vehicles have priority when passing through the intersection, but they may face delay due to the influence of social vehicles. Intersection delay time $d_v$ can be divided into two parts—the dissipation time of vehicles driving in front of rescue vehicles and the time of rescue vehicles passing through the intersection [35]:

$$d_v = d_v^0 + \psi_v \tag{2}$$

where $d_v^0$ refers to the time when rescue vehicles pass through the intersection without delay, and $\psi_v$ represents the dissipation time of other vehicles driving in front of rescue vehicles, which is related to dissipation speed, queuing length and driving distance of the queuing vehicles. Since the arrival rate of traffic flow at the intersection is a random distribution, $\psi_v$ can be considered as a random variable.

#### 2.2.2. Travel Time of a Rescue Route

A rescue route shall be a simple route, and its travel time can be expressed by the travel time of each link and intersection included in the route. The travel time of a rescue route $k$ can be formulated as follows.

$$t2_{k_{rs}} = t1_{ir} + \sum_{e_{mn} \in E} t_{mn} \delta_{mn}^{k_{rs}} + \sum_{v \in V} d_v \delta_{v}^{k_{rs}} \tag{3}$$

Assuming that the travel time of each link and intersection is independent of each other, the expected value and standard deviation of the travel time for route $k$ can be expressed as follows:

$$E(t2_{k_{rs}}) = \sum_{e_{mn} \in E} t_{mn}^0 \delta_{mn}^{k_{rs}} + \sum_{v \in V} d_v^0 \delta_{mn}^{k_{rs}} + E(t1_{ir}) + \sum_{e_{mn} \in E} E(\xi_{mn}) \delta_{mn}^{k_{rs}} + \sum_{v \in V} E(\psi_v) \delta_v^{k_{rs}} \quad (4)$$

$$D(t2_{k_{rs}}) = D(t1_{ir}) + \sum_{e_{mn} \in E} D(\xi_{mn}) \delta_{mn}^{k_{rs}} + \sum_{v \in V} D(\psi_v) \delta_{mn}^{k_{rs}} \quad (5)$$

where $E(\cdot)$ and $D(\cdot)$ are the expected value and standard deviation of random variables, respectively.

According to the central limit theorem, the travel time of a rescue route can be considered to obey the normal distribution with the expected value of $E(t2_{k_{rs}})$ and standard deviation of $D(t2_{k_{rs}})$ [36,37].

$$t2_{k_{rs}} \sim N\left( E(t2_{k_{rs}}), D(t2_{k_{rs}})^2 \right) \quad (6)$$

Meanwhile, the probability density function for the travel time of a rescue route is as follows.

$$\phi(t) = \frac{1}{\sqrt{2\pi} D(t2_{k_{rs}})} \int_{-\infty}^{t} \exp\left( -\frac{(t - E(t2_{k_{rs}}))^2}{2D(t2_{k_{rs}})^2} \right) dt \quad (7)$$

5. Route travel-time reliability

According to the travel characteristics of dangerous goods accident rescue, the travel time reliability of a rescue route is the probability that the rescue vehicle can reach the accident location within the maximum tolerable time, which can be expressed as follows.

$$R_{k_{rs}} = P\{t2_{k_{rs}} \leq RT_i\} = \phi(RT_i) \quad (8)$$

Generally, the more routes that can be selected between the rescue center and the accident location, the greater the possibility that the rescue vehicle reaches the accident location within the maximum tolerable time, and the higher the reliability. If there are multiple routes between the rescue center and the accident location, it can be considered as a parallel system, and the multi-route travel time reliability can be formulated as follows [38].

$$R_{rs} = 1 - \prod_{k_{rs} \in K_{rs}} (1 - R_{k_{rs}}) \quad (9)$$

*2.3. The Optimization Model*

1. Analysis of constraints

The sum of emergency materials, equipment or personnel provided by each rescue center shall meet the demands of the accident location.

$$\sum_{r \in R} EM_{ir} \geq ED_i \quad (10)$$

In addition, the number of emergency materials, equipment or personnel provided by each rescue center shall be less than or equal to its rescue capacity.

$$EM_{ir} \leq EN_{ir} \quad (11)$$

The rescue center $r$ can reach the accident location within the maximum tolerable time by selecting the rescue route $k_{rs}$ with confidence level $\alpha$.

$$R_{k_{rs}} = P\{t2_{k_{rs}} \leq RT_i\} \geq \alpha \quad (12)$$

Constraint (13) ensures that route $k_{rs}$ must not have a loop.

$$\sum_{v \in V} \delta_v^{k_{rs}} \leq 1 \tag{13}$$

The flow balance constraint for each node in a transportation model network is as follows.

$$\sum_{n \in V: e_{mn} \in E} \delta_{mn}^{k_{rs}} - \sum_{n \in V: e_{nm} \in E} \delta_{nm}^{k_{rs}} = \begin{cases} 1, & m = s \\ -1, & m = r \\ 0, & \text{otherwise} \end{cases} \tag{14}$$

2. The objective function

For accident rescue, the arrival time of emergency materials, equipment and personnel and the travel-time reliability of the rescue route are the main concerns. Once a dangerous goods accident occurs, all rescue centers will respond at once. Therefore, the arrival time can be expressed by the sum of the expected value of the route travel time of emergency materials, equipment and personnel of each rescue center.

$$Z_1 = \sum_{i \in I} \sum_{r \in R} [EM_{ir} \cdot E(t2_{k_{rs}})] \tag{15}$$

In terms of reliability, the total reliability of various emergency materials, equipment and personnel arriving at the accident location is used to represent the reliability of the rescue centers relative to the accident location, which can be expressed as follows.

$$Z_2 = \sum_{i \in I} \sum_{r \in R} (EM_{ir} \cdot R_{k_{rs}}) \tag{16}$$

3. The multi-objective rescue resource scheduling model

According to the above analysis, by considering the factors such as the rescue capacity, rescue demand and response time, the multi-objective scheduling model of rescue resources based on travel time reliability is constructed in order to minimize the total arrival time of rescue resources and to maximize total reliability.

$$
\begin{aligned}
& \min Z_1 = \sum_{i \in I} \sum_{r \in R} [EM_{ir} \cdot E(t2_{k_{rs}})], \\
& \max Z_2 = \sum_{i \in I} \sum_{r \in R} (EM_{ir} \cdot R_{k_{rs}}), \\
& \text{s.t.} \\
& \sum_{r \in R} EM_{ir} \geq ED_i, && i \in I, \\
& EM_{ir} \leq EN_{ir}, && i \in I, r \in R, \\
& R_{k_{rs}} = P\{t2_{k_{rs}} \leq RT_i\} \geq \alpha, && i \in I, r \in R, k_{rs} \in K_{rs}, \\
& \sum_{v \in V} \delta_v^{k_{rs}} \leq 1, && k_{rs} \in K_{rs}, \\
& \sum_{n \in V: e_{mn} \in E} \delta_{mn}^{k_{rs}} - \sum_{n \in V: e_{nm} \in E} \delta_{nm}^{k_{rs}} = \begin{cases} 1, & m = s \\ -1, & m = r \\ 0, & \text{otherwise} \end{cases}, && k_{rs} \in K_{rs}, \\
& EM_{ir} \in N, \delta_{mn}^{k_{rs}}, \delta_v^{k_{rs}} \in 0, 1, && i \in I, r \in R, k_{rs} \in K_{rs}
\end{aligned}
\tag{17}
$$

## 3. Solution Algorithms

The rescue resource scheduling problem based on travel time reliability needs to confront two problems: one is the vehicle routing problem from each rescue center to the location of accident under the requirements of reliability, and the other is the quantity allocation of rescue resources provided by each rescue center. In order to reduce the impact caused by the accident as soon as possible, the rescue center with the shortest travel time and highest reliability should be preferred in terms of dispatching rescue resources.

The time used for solving a rescue resource scheduling problem in a realistic scenario may influence the rescue. In order to improve calculation speeds, the solution algorithm for the multi-objective rescue resource scheduling model can be divided into two stages. The first stage is to obtain a set of feasible routes from each rescue center to the location where the accident occurs that meets reliability constraints, and the second stage is to determine the rescue route selected and the quantity of various rescue resources provided by each rescue center.

### 3.1. Obtaining the Set of Feasible Routes

The travel time of the rescue route should meet constraint (12); that is, the rescue route from the rescue unit to the location of accident is bounded. In addition, when determining the vehicle routing problem from the rescue center to the accident location, the travel time and reliability of the route should be considered simultaneously. Therefore, designing a multi-path algorithm with a bound constraint is necessary in order to obtain a set of feasible routes.

1.  Determination of the boundary constraint

The rescue route of each rescue center should meet the constraint of the maximum tolerable time constraint (12). Therefore, according to the definition of travel time reliability of rescue vehicle route $k_{rs}$ (Equation (8)) and the probability density function for route travel time (Equation (7)), the reliability constraint (12) can be transformed into the following:

$$E(t2_{k_{rs}}) + D(t2_{k_{rs}}) \cdot \inf\left\{ G \middle| G = \varphi^{-1}(\alpha) \right\} \leq RT_i \tag{18}$$

where $\varphi^{-1}(\cdot)$ is the inverse function of the distribution function of the standard normal distribution, and $\inf\left\{ G \middle| G = \varphi^{-1}(\alpha) \right\}$ represents the infimum of function $\varphi^{-1}(\alpha)$.

Thus, the boundary constraint of a rescue route can be determined by Equation (18).

2.  Multi-path algorithm

Dijkstra's algorithm is an effective shortest path algorithm, which calculates many nodes during the process of solving the shortest path. If the label information of these nodes is retained, it is conducive to the calculation of multi-path.

The rescue route studied in this paper is a simple path with a boundary constraint. Therefore, it is necessary to meet the boundary constraints when selecting temporary label nodes. Thus, this paper provides the definition of qualified nodes firstly.

**Definition 1.** *Let* $t2_{p_{rm}}$ *represent the travel time of route* $p_{rm}$ *from the rescue center* r *to a node* $v_m$, *and* $v_n$ *is the adjacent node of* $v_m$. *If* $v_n$ *is not on route* $p_{rm}$ *and the travel time from the rescue center* r *to a node* n *satisfies the following:*

$$E(t2_{p_{rm}} + t_{mn}) + D(t2_{p_{rm}} + t_{mn}) \cdot \inf\left\{ K \middle| K = \varphi^{-1}(\alpha) \right\} \leq RT_i \tag{19}$$

*then* $v_n$ *is called the qualified node for route* $p_{rm}$.

Definition 1 ensures that the generated route is a simple path and meets the bound constraint. Based on Dijkstra's algorithm and the definition of qualified nodes, a multi-path algorithm can be designed by retaining the temporary information of nodes. For convenience, $p_{k_{rs}}$ is used to represent the $k_{rs}$-th route from the rescue center $r$ to the accident location $s$, and $T(\cdot)$ and $P(\cdot)$ represent temporary label and permanent label, respectively.

The specific steps of the multi-path algorithm are as follows:

*   Step 1: Let the number of routes $k_{rs}$ = 1, the temporary label of other notes ($v_m \in V \backslash r$)$T(v_m) = +\infty$, the departure time of the rescue center $T(r) = t1_{ir}$, $p_{k_{rs}} = r$, and let node $r$ be a permanent label.
*   Step 2: Select the smallest permanent label $v_i$ and check whether each adjacent node $v_j$ of $v_i$ is a qualified note according to Equation (18). If it is a qualified node, add the

temporary label of $v_j$ and $T(v_j) = T(v_i) + d_i + t_{ij}$ and proceed to Step 3. If there is no qualified node, let $P(v_i) = +\infty$, and proceed to Step 3.

- Step 3: Select the node $v_n$ with the smallest expected value among all the temporary labels, and mark it as the permanent label; let $P(v_n) = \min_j \{ E[T(v_j)] \}$, $p_{k_{rs}} = p_{k_{rs}} \cup n$.

- Step 4: If $v_n$ is the location of accident, add route $p_{k_{rs}}$ to $K_{k_{rs}}$ and mark the permanent label value of nodes on this route with $+\infty$, $k_{rs} = k_{rs} + 1$; then, proceed to Step 3. Otherwise, proceed to Step 2.

For the routes in the calculated route set $K_{k_{rs}}$, the expected values of travel time and reliability are calculated according to Equations (7), (8) and (12). Then, the dominant path is eliminated, and the rescue route set between each rescue center $r$ and the accident location $s$ can be obtained.

### 3.2. Determining Rescue Route and Rescue Resource Supply

Since the model in this paper is a bi-objective programming model, the expected value and the reliability of the route travel time should be considered when determining rescue route selection and the supply of rescue resources in this stage. One of the widely used algorithms among metaheuristics proposed to solve multi-objective APP problems is the Pareto-based non-dominated sorting genetic algorithm called NSGA-II (Non-dominated Sorting Genetic Algorithm-II) [39]. Therefore, in this stage, the NSGA-II algorithm is used to determine the scheduling of rescue resources for each rescue center.

#### 3.2.1. Chromosome Structure

In this stage, it is necessary to determine the rescue route of each rescue center and the supply of rescue resources by each rescue center.

Accordingly, the index of rescue routes $k_{rs}$ and supply quantity of rescue centers $EM_{ir}$, which are the decision variables, are both positive integers. Therefore, integer encoding is adopted, and $CH = (EM, k)$ is designed as a chromosome. For sub-chromosome EM, a positive integer $I \times R$ matrix is used to represent the supply quantity of rescue centers. A gene in this sub-chromosome represents the supply quantity ($EM_{ir}$) of the rescue center $r$ for rescue resource $i$. For sub-chromosome k, a vector with $r$ elements is adopted to represent the index of rescue routes for each rescue center. Figure 2 shows a chromosome with $I$ rescue resource types and $R$ rescue center.

| Rescue center (*r*) | 1 | 2 | ... | *R* |
|---|---|---|---|---|
| Resource type 1 | $EM_{11}$ | $EM_{12}$ | ... | $EM_{1R}$ |
| Resource type 2 | $EM_{21}$ | $EM_{22}$ | ... | $EM_{2R}$ |
| ... | | ... | | |
| Resource type  *I* | $EM_{I1}$ | $EM_{I2}$ | ... | $EM_{IR}$ |

EM

| k | Route ( $k_{rs}$ ) | $k_{1s}$ | $k_{2s}$ | ... | $k_{Rs}$ |
|---|---|---|---|---|---|

**Figure 2.** A simple example of a chromosome.

#### 3.2.2. Initialization

In the process of initial population generation, the genes of sub-chromosomes EM need to meet constraints (10) and (11). In order to improve the quality of the initial population, rescue centers are sorted from the perspective of two objective functions.

Let $o^1_{ir}$ denote the sequence number of the rescue center $r$ sorted in descending order according to the minimum expected travel time of the rescue route for rescue resource $i$. Correspondingly, let $o^2_{ir}$ represent the sequence number of the rescue center $r$ sorted in ascending order according to the reliability of the rescue route for rescue resource $i$. Generally, the higher the amount of resources provided by the top rescue centers, the

better the quality of chromosomes for this object. Let us set the comprehensive ranking value of rescue centers for rescue resource $i$ as $o_{ir}$ and the weight of random sorting as $\omega_i$, respectively. Then, comprehensive ranking can be defined as follows.

$$o_{ir} = \omega_i \cdot o_{ir}^1 + (1 - \omega_i) \cdot o_{ir}^2 \qquad (20)$$

For sub-chromosome k, the first stage of this algorithm has obtained the set of feasible routes between each rescue center and the accident location, and at this stage, we only need to select an appropriate route.

The process of generating a chromosome based on random sorting is as follows:

- Step 1: Parameter setting: set $R$, $I$, $k_{rs}$, $o_{ir}^1$, $o_{ir}^2$, $ED_i$ and $EN_{ir}$. Let $g = 1$ and $\omega_i = 0$.
- Step 2: For $g = 1$ to *popsize* (the number of chromosomes), repeat Step 3 to Step 5.
- Step 3: For $i = 1$ to $I$, calculate the comprehensive ranking value $o_{ir}$ and sort the rescue centers according to $o_{ir}$ in descending order.
- Step 4: For $r = 1$ to $R$, arrange the supply of rescue resource $i$ in the order of $o_{ir}$: let the supply of the first rescue center be $EM_{ir} = \min\{EN_{ir}, ED_i\}$, $ED_i = ED_i - \min\{EN_{ir}, ED_i\}$ and calculate the supply of the subsequent rescue center $EM_{ir} = \min\{EN_{ir}, ED_i\}$. If $ED_i = 0$, set $\omega_i = \omega_i + 1/(popsize - 1)$ and proceed to Step 3.
- Step 5: For $r = 1$ to $R$, generate a random integer number $k_r \in [1, k_{rs}]$ as a gene of sub-chromosome k.

For example, if the demand for the rescue resources $ED_i$ ($I = 3$) is (50, 40, 40), the rescue capacity $EN_{ir}$ and the comprehensive ranking value $o_{ir}$ of the rescue centers ($r = 4$) are shown in Figure 3, and the sub-chromosome of the supply quantity for rescue resources will be generated according to the above chromosome generation method. For resource 1, according to the comprehensive ranking value $o_{1r}$, the first place is rescue center 1, and its supply quantity can be determined as min {50, 20} = 20. The second place is rescue center 2, and it can be determined that the supply quantity is min {50–20, 10} = 10. Similarly, the third place is rescue center 4, and its supply quantity is min {50–20–10, 30} = 20. The demand of resource 1 has been met, and we can obtain a feasible sub-chromosome of the supply quantity by repeating this operation, as shown in Figure 3.

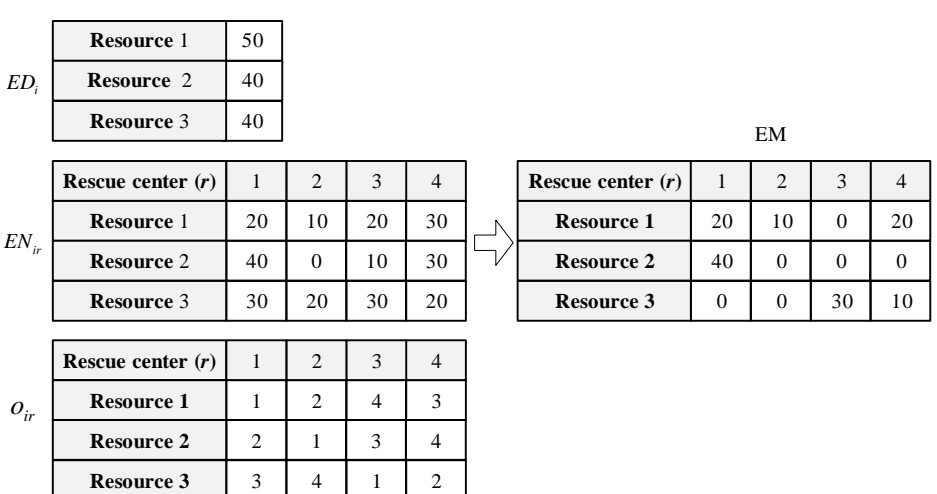

**Figure 3.** Illustration of sub-chromosome EM generation.

Obviously, the above chromosome generation method satisfies constraints (10) and (11).

### 3.2.3. Crossover Operator

The crossover operation for sub-chromosome EM is carried out by a linear combination of two parent chromosome genes. Let us randomly select two parent chromosomes and randomly select a rescue resource type for crossover according to the following equation:

$$EM1'_{ir} = \lambda EM1_{ir} + (1 - \lambda)EM2_{ir},$$
$$EM2'_{ir} = (1 - \lambda)EM1_{ir} + \lambda EM2_{ir}, \quad \lambda \in (0,1) \tag{21}$$

where $EM1_{ir}$ and $EM2_{ir}$ are the genes of the supply quantity for selected parents, and $EM1'_{ir}$ and $EM2'_{ir}$ are the genes of an offspring. Figure 4 shows the crossover operation of resource 1 for sub-chromosome EM.

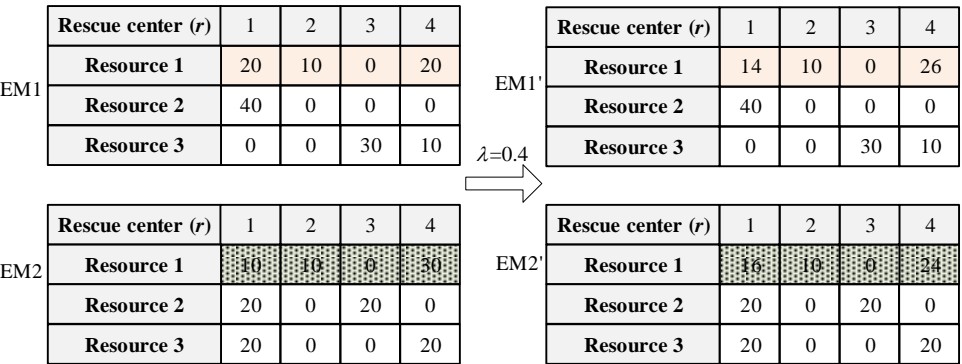

**Figure 4.** Illustration of crossover for sub-chromosome EM.

For sub-chromosome k, generate a random integer number $r \in [1, R]$ and exchange genes in location $r$ of two parent sub-chromosomes. The crossover process for sub-chromosome k is illustrated in Figure 5.

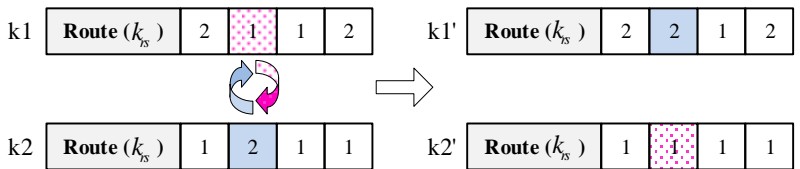

**Figure 5.** Illustration of crossover for sub-chromosome k.

### 3.2.4. Mutation Operator

For sub-chromosome EM, the mutation operation is carried out by regenerating the random sorting weight. Let us randomly select a rescue resource type $i$, generate a random sorting weight $\omega_i$, calculate the comprehensive ranking value $o_{ir}$ and regenerate the sub-chromosome EM of $i$ according to the generation method based on random sorting. Assume that the resource 2 is selected for mutation. Figure 6 shows the mutation process for sub-chromosome EM.

For sub-chromosome k, let us randomly select a gene and regenerate this gene in the feasible region.

### 3.2.5. Selection Process

In this paper, non-dominated sorting and crowding distance-based fitness assignment strategies of NSGA-II algorithm are used to implement competitive selection [40].

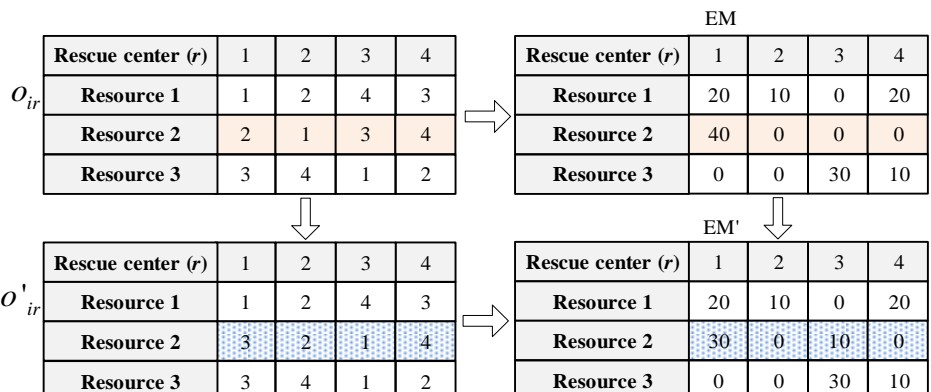

**Figure 6.** Illustration of mutation for sub-chromosome EM.

## 4. Experimental Analysis

A regional road network is shown in Figure 7. The accident location $s = 1$ and points 2–7 are rescue centers. Table 4 shows the resource demand of the accident location and the corresponding capacity of each rescue center. The maximum tolerable transport time for rescue resources and the assembly time of rescue resources for each rescue center are shown in Table 5. The travel time of links is listed in Table 6. In addition, the time for rescue vehicles to pass through the intersection without delay $d_v^0 = 0.05$ min and the dissipation time of other vehicles at the intersection are shown in Table 7.

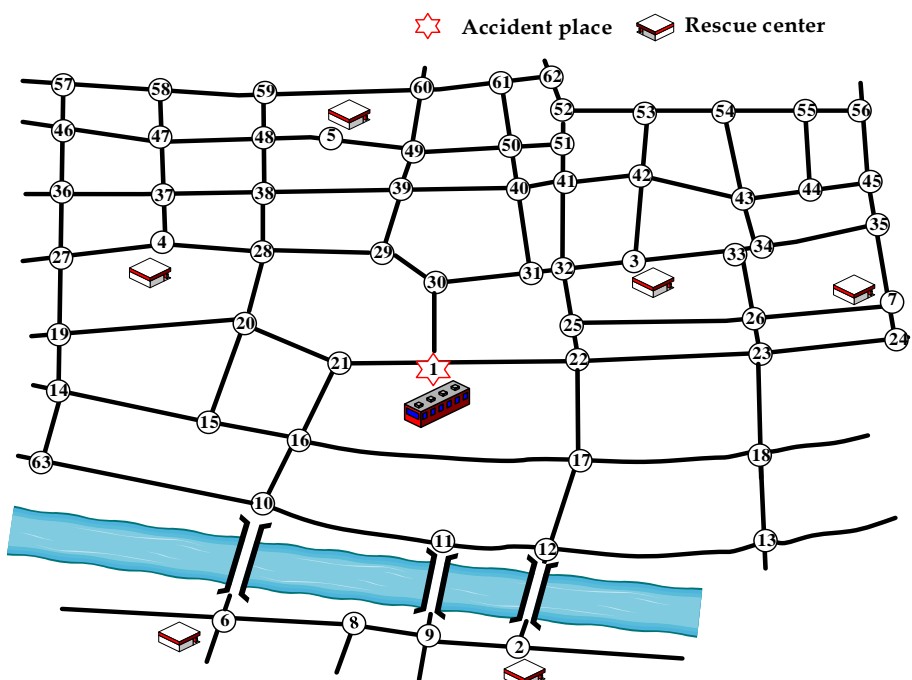

**Figure 7.** Schematic diagram of a regional road network.

**Table 4.** Demand for rescue resources and rescue capacity of rescue centers.

| Rescue Resource $i$ | $ED_i$ | $EN_{ir}$ | | | | | |
|---|---|---|---|---|---|---|---|
| | | **2** | **3** | **4** | **5** | **6** | **7** |
| 1 | 120 | 80 | 60 | 50 | 0 | 0 | 70 |
| 2 | 100 | 70 | 50 | 60 | 60 | 0 | 0 |
| 3 | 80 | 0 | 0 | 50 | 50 | 50 | 0 |
| 4 | 90 | 50 | 0 | 0 | 0 | 70 | 50 |

**Table 5.** The maximum tolerable transport time and the assembly time of each rescue center.

| Rescue Resource $i$ | $RT_i$/min | $t1_{ir}$/min $(E(\cdot),D(\cdot))$ | | | | | |
|---|---|---|---|---|---|---|---|
| | | 2 | 3 | 4 | 5 | 6 | 7 |
| 1 | 15 | (3.5, 1.5) | (3.5, 1.2) | (4, 1) | n.a. | n.a. | (3.5, 1.5) |
| 2 | 15 | (4, 1.5) | (4, 1.5) | (5, 1.2) | (4.5, 1.5) | n.a. | n.a. |
| 3 | 10 | n.a. | n.a. | (2, 0.8) | (1.5, 0.5) | (1.5, 0.9) | n.a. |
| 4 | 10 | (1.5, 0.8) | n.a. | n.a. | n.a. | (2, 1.2) | (1.5, 1) |

**Table 6.** The travel time of links.

| $m,n$ | $t^0_{mn}$/min | $\xi_{mn}$/min $(E(\cdot),D(\cdot))$ | $m,n$ | $t^0_{mn}$/min | $\xi_{mn}$/min $(E(\cdot),D(\cdot))$ |
|---|---|---|---|---|---|
| 2, 9 | 0.6 | (0.5, 0.05) | 7, 26 | 0.8 | (0.5, 0.1) |
| 8, 9 | 0.5 | (0.3, 0.05) | 7, 24 | 0.4 | (0.1, 0.05) |
| 6, 8 | 0.8 | (0.5, 0.1) | 7, 35 | 0.8 | (0.5, 0.1) |
| 6, 10 | 0.6 | (0.5, 0.05) | 4, 28 | 0.8 | (0.5, 0.1) |
| 2, 12 | 0.7 | (0.5, 0.08) | 28, 29 | 1.0 | (0.6, 0.1) |
| 9, 11 | 0.6 | (0.5, 0.05) | 29, 30 | 0.5 | (0.3, 0.05) |
| 11, 12 | 0.7 | (0.5, 0.08) | 30, 31 | 0.7 | (0.5, 0.08) |
| 10, 11 | 1.2 | (0.7, 0.1) | 31, 32 | 0.3 | (0.1, 0.05) |
| 10, 16 | 0.8 | (0.5, 0.1) | 3, 32 | 0.4 | (0.1, 0.05) |
| 12, 17 | 0.9 | (0.55, 0.1) | 3, 33 | 0.7 | (0.5, 0.08) |
| 16, 17 | 2.0 | (0.8, 0.1) | 4, 37 | 0.5 | (0.3, 0.05) |
| 16, 21 | 0.7 | (0.5, 0.08) | 37, 38 | 0.7 | (0.5, 0.08) |
| 17, 22 | 0.8 | (0.5, 0.1) | 28, 38 | 0.8 | (0.5, 0.1) |
| 17, 18 | 1.0 | (0.6, 0.1) | 38, 39 | 1.0 | (0.6, 0.1) |
| 15, 16 | 0.7 | (0.5, 0.08) | 38, 48 | 0.5 | (0.3, 0.05) |
| 15, 20 | 1.0 | (0.6, 0.1) | 5, 48 | 0.4 | (0.1, 0.05) |
| 18, 23 | 0.8 | (0.5, 0.1) | 5, 49 | 0.6 | (0.5, 0.05) |
| 20, 21 | 0.8 | (0.5, 0.1) | 39, 49 | 0.4 | (0.1, 0.05) |
| 1, 21 | 0.6 | (0.5, 0.05) | 39, 40 | 0.5 | (0.3, 0.05) |
| 1, 22 | 0.8 | (0.5, 0.1) | 49, 50 | 0.5 | (0.3, 0.05) |
| 22, 23 | 1.0 | (0.6, 0.1) | 29, 39 | 0.4 | (0.1, 0.05) |
| 23, 24 | 0.8 | (0.5, 0.1) | 31, 40 | 0.8 | (0.5, 0.1) |
| 20, 28 | 0.5 | (0.3, 0.05) | 40, 50 | 0.4 | (0.1, 0.05) |
| 1, 30 | 0.7 | (0.5, 0.08) | 40, 41 | 0.3 | (0.1, 0.05) |
| 22, 25 | 0.3 | (0.1, 0.05) | 50, 51 | 0.4 | (0.1, 0.05) |
| 25, 32 | 0.5 | (0.3, 0.05) | 41, 51 | 0.3 | (0.1, 0.05) |
| 25, 26 | 1.0 | (0.6, 0.1) | 32, 41 | 0.9 | (0.55, 0.1) |
| 26, 33 | 0.6 | (0.5, 0.05) | 33, 34 | 0.2 | (0.1, 0.05) |
| 23, 26 | 0.4 | (0.1, 0.05) | 34, 35 | 0.7 | (0.5, 0.08) |
| 37, 47 | 0.5 | (0.3, 0.05) | 50, 61 | 0.7 | (0.5, 0.08) |
| 47, 48 | 0.7 | (0.5, 0.08) | 51, 52 | 0.3 | (0.1, 0.05) |
| 48, 50 | 0.4 | (0.1, 0.05) | 51, 42 | 0.8 | (0.5, 0.1) |
| 49, 60 | 0.7 | (0.5, 0.08) | 42, 43 | 0.6 | (0.5, 0.08) |

**Table 7.** The dissipation time of other vehicles at the intersection.

| $v$ | $\psi_v$/min $(E(\cdot),D(\cdot))$ | $v$ | $\psi_v$/min $(E(\cdot),D(\cdot))$ | $v$ | $\psi_v$/min $(E(\cdot),D(\cdot))$ |
|---|---|---|---|---|---|
| 6 | (0.15, 0.05) | 22 | (0.15, 0.05) | 35 | (0.1, 0.03) |
| 8 | (0.1, 0.03) | 23 | (0.1, 0.03) | 37 | (0.1, 0.03) |
| 9 | (0.1, 0.03) | 24 | (0.1, 0.03) | 38 | (0.15, 0.05) |
| 10 | (0.2, 0.05) | 25 | (0.15, 0.05) | 39 | (0.15, 0.05) |
| 11 | (0.15, 0.05) | 26 | (0.1, 0.03) | 40 | (0.15, 0.05) |
| 12 | (0.2, 0.05) | 28 | (0.15, 0.05) | 41 | (0.1, 0.03) |
| 15 | (0.1, 0.03) | 29 | (0.15, 0.05) | 47 | (0.15, 0.05) |
| 16 | (0.15, 0.05) | 30 | (0.15, 0.05) | 48 | (0.15, 0.05) |
| 17 | (0.15, 0.05) | 31 | (0.2, 0.05) | 49 | (0.25, 0.15) |
| 18 | (0.1, 0.03) | 32 | (0.25, 0.15) | 50 | (0.15, 0.05) |
| 20 | (0.1, 0.03) | 33 | (0.1, 0.03) | 51 | (0.15, 0.05) |
| 21 | (0.15, 0.05) | 34 | (0.1, 0.03) | 52 | (0.1, 0.03) |

Let us set the confidence level $\alpha = 0.9$ by using the multi-path algorithm with the bound constraint; the experiment is then performed on DELL Vostro 3800-R1846 (Intel

Core i3 4130 (3.4 GHz 8 GB memory)) for 10 times, and the average runtime is about 10.3 s. The set of feasible routes from each rescue center to the location where the accident occurs can be obtained, which is listed in Tables 8–11. Then, the dominant path is eliminated, and the rescue route set between each rescue center *r* and the accident location can be obtained.

**Table 8.** Feasible routes and their reliability for rescue resource 1.

| Rescue Center (*r*) | Route | $t2_{k_{rs}}$/min ($E(\cdot),D(\cdot)$) | $R_{k_{rs}}$ |
|---|---|---|---|
| 2 | 2-12-17-22-1 | (9.4, 2.03) | 0.994 |
| | 2-9-11-12-17-22-1 | (11.95, 2.21) | 0.916 |
| | 2-9-11-10-16-21-1 | (11.7, 2.16) | 0.937 |
| | 2-12-17-16-21-1 | (11.6, 2.11) | 0.947 |
| 3 | 3-32-25-22-1 | (7.2, 1.7) | 1 |
| | 3-32-31-30-1 | (7.55, 1.71) | 1 |
| | 3-33-26-23-22-1 | (9.85, 1.72) | 0.999 |
| | 3-33-26-25-22-1 | (9.8, 1.74) | 0.999 |
| 4 | 4-28-20-21-1 | (9.05, 1.43) | 1 |
| | 4-28-29-30-1 | (9.5, 1.47) | 1 |
| | 4-37-38-39-29-30-1 | (11.05, 1.64) | 0.992 |
| | 4-37-38-28-20-21-1 | (11.4, 1.64) | 0.986 |
| 7 | 7-26-25-22-1 | (8.65, 1.98) | 0.999 |
| | 7-24-23-22-1 | (8.7, 1.96) | 0.999 |
| | 7-26-25-32-31-30-1 | (11.1, 2.29) | 0.956 |
| | 7-35-34-33-32-25-22-1 | (11.65, 2.4) | 0.919 |

**Table 9.** Feasible routes and their reliability for rescue resource 2.

| Rescue Center (*r*) | Route | $t2_{k_{rs}}$/min ($E(\cdot),D(\cdot)$) | $R_{k_{rs}}$ |
|---|---|---|---|
| 2 | 2-12-17-22-1 | (9.9, 2.03) | 0.994 |
| | 2-9-11-10-16-21-1 | (12.2, 2.16) | 0.903 |
| | 2-12-17-16-21-1 | (12.1, 2.11) | 0.915 |
| 3 | 3-32-25-22-1 | (7.6, 1.9) | 1 |
| | 3-32-31-30-1 | (7.95, 1.91) | 1 |
| | 3-33-26-23-22-1 | (10.35, 2.02) | 0.989 |
| | 3-33-26-25-22-1 | (10.3, 2.04) | 0.989 |
| 4 | 4-28-20-21-1 | (10.05, 1.63) | 0.999 |
| | 4-28-29-30-1 | (10.5, 1.67) | 0.996 |
| | 4-37-38-39-29-30-1 | (12.05, 1.84) | 0.946 |
| | 4-37-38-28-20-21-1 | (12.4, 1.84) | 0.921 |
| 5 | 5-49-39-29-30-1 | (9.1, 2.08) | 0.998 |
| | 5-48-38-29-20-21-1 | (11.25, 2.13) | 0.961 |
| | 5-49-50-40-31-30-1 | (11.85, 2.26) | 0.918 |
| | 5-48-38-28-29-30-1 | (11.7, 2.18) | 0.935 |

**Table 10.** Feasible routes and their reliability for rescue resource 3.

| Rescue Center (*r*) | Route | $t2_{k_{rs}}$/min ($E(\cdot),D(\cdot)$) | $R_{k_{rs}}$ |
|---|---|---|---|
| 4 | 4-28-20-21-1 | (7.05, 1.23) | 0.992 |
| | 4-28-29-30-1 | (7.49, 1.27) | 0.976 |
| 5 | 5-49-39-29-30-1 | (6.5, 1.08) | 0.999 |
| | 5-48-38-29-20-21-1 | (8.25, 1.13) | 0.939 |
| 6 | 6-10-16-21-1 | (6.85, 1.33) | 0.991 |

**Table 11.** Feasible routes and their reliability for rescue resource 4.

| Rescue Center (*r*) | Route | $t2_{k_{rs}}$/min ($E(\cdot)$,$D(\cdot)$) | $R_{k_{rs}}$ |
|---|---|---|---|
| 2 | 2-12-17-22-1 | (7.4, 1.33) | 0.975 |
| 6 | 6-10-16-21-1 | (7.35, 1.63) | 0.948 |
| 7 | 7-26-25-22-1 | (6.65, 1.48) | 0.988 |
| | 7-24-23-22-1 | (6.7, 1.46) | 0.988 |

Based on the rescue route set obtained by the first stage, the NSGA-II algorithm is used to determine the scheduling of rescue resources for each rescue center in the second stage. The size of population is 20, the maximum number of iterations is 100, the crossover probability is $P_c = 0.5$ and the mutation probability is $P_m = 0.3$. Similarly, the algorithm is executed 10 times in this stage, with an average runtime of about 5 s. The Pareto solutions are shown in Figure 8. In the obtained Pareto solution set, two schemes with the minimum expected total travel time (Scheme 1) and the maximum total reliability (Scheme 7) are selected for analysis, and the results of rescue resource scheduling for the two schemes are shown in Table 12. The object values of Scheme 1 are 2943 and 386.84, and the object values of Scheme 7 are 3018.5 and 388.05. Compared with Scheme 1, the reliability of Scheme 7 is improved, but the total expected travel time increased. Rescue resource 1 of Scheme 7 needs to dispatch resources from three centers due to the capacity limitation of rescue center 4, while Scheme 1 only needs to dispatch resources from two centers.

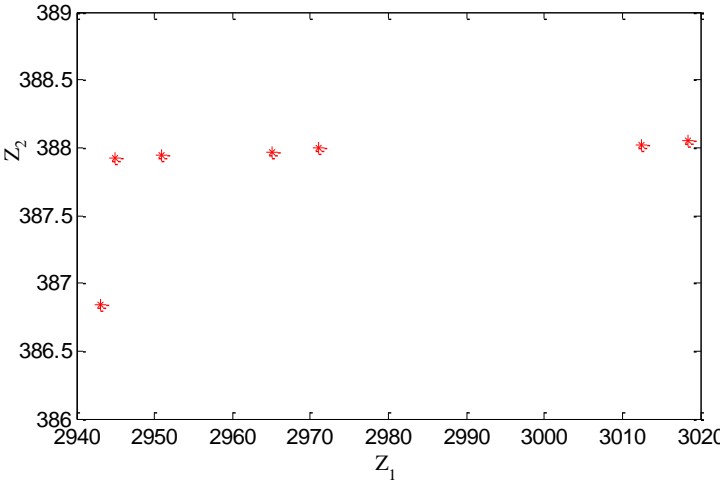

**Figure 8.** The distribution of obtained Pareto solutions.

**Table 12.** The rescue resource scheduling of Scheme 1 and Scheme 7.

| Rescue Resource *i* | $ED_i$ | $EM_{ir}$ | | | | | | | | |
|---|---|---|---|---|---|---|---|---|---|---|
| | | Scheme 1 | | | | Scheme 7 | | | | |
| | | 3 | 5 | 6 | 7 | 2 | 3 | 4 | 5 | 7 |
| 1 | 120 | 60 | 0 | 0 | 60 | 0 | 60 | 50 | 0 | 10 |
| 2 | 100 | 50 | 50 | 0 | 0 | 0 | 50 | 50 | 0 | 0 |
| 3 | 80 | 0 | 50 | 30 | 0 | 0 | 0 | 30 | 50 | 0 |
| 4 | 90 | 0 | 0 | 40 | 50 | 40 | 0 | 0 | 0 | 50 |

By using the method of information entropy, synthesized objective values can be obtained, and the synthesized objective values of Scheme 2 and Scheme 3 exceed 0.9. It can be observed that Scheme 2 is relatively better in two objectives (2945, 387.92), and the rescue resource scheduling of Scheme 2 is shown in Figure 9.

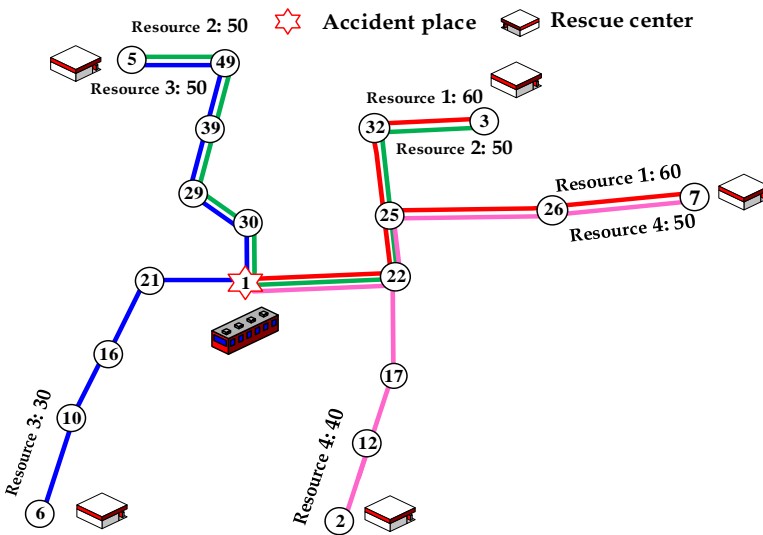

**Figure 9.** The rescue resource scheduling of Scheme 2.

## 5. Conclusions

The accident rescue of railway dangerous goods accident has the characteristic of time urgency, the accident itself has continuous expansion, the traffic environment has uncertainty and the accident rescue needs multiple rescue resources. Therefore, it is necessary to integrate and reasonably arrange appropriate rescue resource scheduling and routing plans. According to the analysis of travel time and reliability for rescue routes, considering the factors such as rescue capacity, rescue demand and response time, a multi-objective scheduling model of rescue resources based on travel time reliability is constructed in order to minimize the total arrival time of rescue resources and to maximize total reliability. Considering travel time and reliability of the rescue route, the proposed model is more reliable than the traditional model.

Furthermore, a two-stage algorithm is designed to solve this problem. In the first stage, a multi-path algorithm with a bound constraint is used to obtain the set of feasible routes from each rescue center relative to the accident location. In the second stage, the NSGA-II algorithm is used to determine the scheduling of rescue resources for each rescue center. Finally, a regional road network is collected to test multi-objective scheduling and algorithm. The results show that the designed two-stage algorithm can simplify problem solving and effectively improve solving speed, and it is valid for solving the rescue resource scheduling problem of dangerous goods accidents.

In order to reduce emergency response time when an accident occurs, the location in relation to transport and storage of dangerous goods should formulate an emergency plan. At the same time, emergency personnel should be trained to improve their emergency safety technical knowledge and master certain abilities in order to prevent and deal with accidents. Emergency drills should be organized regularly to improve risk awareness in dealing with emergencies and abilities related to emergency response. Furthermore, each rescue unit should reserve sufficient materials required for railway dangerous goods accidents within the emergency scope and plan the rescue route in advance.

In this paper, we assumed that the demand for the rescue resources is known. In reality, the running time may be uncertain due to the complexity of dangerous goods accidents. Moreover, travel times are time-varying and stochastic in transportation because of the effects of traffic flow and weather. Therefore, future research can consider the following:

1. Considering the uncertainty of the demand for rescue resources to design a model and algorithm of rescue resource scheduling problem;
2. Considering the travel time of links in stochastic, time-varying transportation networks to research the rescue resource scheduling problem;

3. Considering the priority of rescue equipment and resources to research on the rescue resource scheduling problem.

**Author Contributions:** The contributions of the authors were distributed in the following manner: Conceptualization, L.L.; methodology, X.Y. and L.L.; software, X.Y.; validation, L.L.; formal analysis, X.Y.; investigation, L.L.; data curation, X.Y.; writing—original draft preparation, X.Y.; writing—review and editing, L.L; supervision, X.Y.; project administration, X.Y.; funding acquisition, X.Y. and L.L. All authors have read and agreed to the published version of the manuscript.

**Funding:** This work is supported by the National Natural Science Foundation of China (Grant No. 71761024), the Natural Science Foundation of Gansu Province of China (No. 21JR1RA236) and Double-First Class Major Research Programs, Educational Department of Gansu Province (No. GSSYLXM-04).

**Institutional Review Board Statement:** Not applicable.

**Informed Consent Statement:** Not applicable.

**Data Availability Statement:** Data sharing is not applicable to this article.

**Conflicts of Interest:** The authors declare no conflict of interest.

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
