# Peer review of "Travel Time Reliability-Based Rescue Resource Scheduling for Accidents Concerning Transport of Dangerous Goods by Rail"

_algorithms, doi:10.3390/a14110325_

Round 1

Reviewer 1 Report

In the paper titled “Research on rescue resource scheduling of railway dangerous goods accidents based on Reliability" the authors propose a multi-objective scheduling model of rescue resources based on travel time reliability to minimize the total arrival time of rescue resources and maximize the total reliability.

So, the objective of the paper is relevant for the journal.

The manuscript topics fit well to the journal scope.

In my opinion, the literature review reported in the section Introduction (I) is adequate to outline the objectives of the proposed research.

A greater effort is required from the authors to highlight the main results of the proposed research. For example, a sentence could be added to the abstract to underline the best result achieved (for example, the proposed model is more reliable than the other models because of ..., or the results show that ...). Similarly, in the conclusions, it is necessary to better highlight the contribution provided by this research in the scientific panorama. From reading the article, it is not yet possible to deduce the contribution provided by the research.

Author Response

Point 1: A greater effort is required from the authors to highlight the main results of the proposed research. For example, a sentence could be added to the abstract to underline the best result achieved (for example, the proposed model is more reliable than the other models because of ..., or the results show that ...). Similarly, in the conclusions, it is necessary to better highlight the contribution provided by this research in the scientific panorama. From reading the article, it is not yet possible to deduce the contribution provided by the research.

Response 1: Thank you for your valuable comments. We updated the manuscript by adding corresponding description in the sections of abstract and conclusions. Thank you again for your positive comments and valuable suggestions to improve the quality of our manu

Reviewer 2 Report

The paper provides a multi-objective scheduling model of rescue resources 
based on travel time reliability and a two-stage algorithm is designed to solve this problem. The paper is interesting and generally well written. However, some issues need to be solved:

  1. The title of the paper is a little bit confusing, mainly the part "based on reliability" since not only reliability is taken into consideration. Moreover, "reliability" denotes a general concept and this paper employs only a special type of reliability (travel time reliability). Maybe the authors will find a more suitable title to describe their work;
  2. The time needed to solve the optimization problem in a realistic scenario may influence the interventions. Presenting such a time evaluation may be beneficial for the reader.
  3. Line 65: it is written "develops" instead of "develop";
  4. Line 82: seems to be a typo there ("resource scheduling resources");
  5. Line 86: the sequence "the time value is much higher than the economic value" is confusing. On what basis the authors wrote this? Please reshape.
  6. Line 333: the sequence "Obviously, the above chromosome generation method obviously" needs to be reshaped ("obviously" is used two times).

Author Response

Point 1: The title of the paper is a little bit confusing, mainly the part "based on reliability" since not only reliability is taken into consideration. Moreover, "reliability" denotes a general concept and this paper employs only a special type of reliability (travel time reliability). Maybe the authors will find a more suitable title to describe their work;

Response 1: Thank you for your suggestion. We changed the title to “Travel time reliability-based rescue resource scheduling for accidents concerning transport of dangerous goods by rail” combined with the suggestions of Reviewer 2.

Point 2: The time needed to solve the optimization problem in a realistic scenario may influence the interventions. Presenting such a time evaluation may be beneficial for the reader.

Response 2: Thank you for your advice. We updated the manuscript by adding corresponding description in Section 3 and Section 4, and used the example to evaluate the runtime of the two-stage algorithm.

Point 3: Line 65: it is written "develops" instead of "develop";

Response 3: Thank you for your reminding.

Point 4: Line 82: seems to be a typo there ("resource scheduling resources");

Response 4: Thank you for your reminding, again.

Point 5: Line 86: the sequence "the time value is much higher than the economic value" is confusing. On what basis the authors wrote this? Please reshape.

Response 5: Thank you for your advice. We updated the manuscript by deleting corresponding comment.

Point 6:  Line 333: the sequence "Obviously, the above chromosome generation method obviously" needs to be reshaped ("obviously" is used two times).

Response 6: Thank you for your reminding. We have corrected it.

Reviewer 3 Report

Page 1, Title of the article “Research on rescue resource scheduling of railway dangerous goods accidents based on Reliability”: My suggestion is to change the title to “Reliability based rescue resource scheduling for accidents concerning transport of dangerous goods by rail”.

Page 1, lines 24-25, “Currently, large quantity of different types of dangerous goods is transported by road, rail, water and air.”. My suggestion is to replace “water” with “inland waterways” (https://unece.org/about-adnand) and “maritime”. My suggestion is also to add “pipelines”.

Section 1. Introduction: My suggestion is to include a paragraph or two which must be dedicated to the Regulation concerning the International Carriage of Dangerous Goods by Rail (RID). Please also include the respective references (e.g., https://otif.org/en/?page_id=1105). I consider that it is very important for the reader to see the definitions and terminology concerning the topic “Dangerous Goods by Rail”.

Section 2. The Reliability Based Rescue Resource Scheduling Model: Please include a Data Flow Chart describing all your methodological steps so that the reader can obtain a clear overview of your work from the early beginning of the manuscript.

Page 2, lines 76-78: There are 16 references which are associated with the specific sentence. My suggestion is to try to enhance the text which is associated with these references since their topics are very interesting.

Page 3, lines 102 and 103, “Therefore, in case of railway dangerous goods accident, it is urgent to transport all kinds of rescue equipment and resources to the accident place.”. My suggestion is to provide analytical details concerning “rescue equipment and resources”. In addition, please justify why “all kinds…” are needed. Is it possible to having prioritization when considering the “rescue equipment and resources” taking into account the available technology for incident management?

Page 6, line 213: Please consider the idea to number the equations presented in this specific part of the manuscript (just above Section 3. Solution Algorithms).

Page 7, line 249: It seems that “Definition” has a different font height (points).

Page 11, Section 4. Experimental analysis: Your experiment considered a regional road network. What would be the differences in your approach in the case of a semi urban road network where the basic traffic characteristics (traffic volume, speed, density) are more difficult?

Page 11, Table 5. The maximum tolerable transport time and the assembly time of each rescue center.: Please insert “n.a.” (not applicable or not available) in the empty cells of the specific Table.

Please change Section “3 Solution Algorithms” to Section “3. Solution Algorithms”.

Please change Section “4 Experimental analysis” to Section “4. Experimental Analysis”.

Please change Section “5 Conclusions” to Section “5. Conclusions”.

Section 5. Conclusions: Please include a summary with the constraints and limitations of your research. In addition, my suggestion is to formulate some policy recommendations arising from your findings and then to address these recommendations to the respective stakeholders (e.g., railway operators, rescue centers etc.). Finally, please include the future directions of the research on the topic of the article.

Author Response

Point 1: Page 1, Title of the article “Research on rescue resource scheduling of railway dangerous goods accidents based on Reliability”: My suggestion is to change the title to “Reliability based rescue resource scheduling for accidents concerning transport of dangerous goods by rail”.

Response 1: Thank you for your suggestion. We changed the title to “Travel time reliability-based rescue resource scheduling for accidents concerning transport of dangerous goods by rail” combined with the suggestions of Reviewer 1.

Point 2: Page 1, lines 24-25, “Currently, large quantity of different types of dangerous goods is transported by road, rail, water and air.”. My suggestion is to replace “water” with “inland waterways” (https://unece.org/about-adnand) and “maritime”. My suggestion is also to add “pipelines”.

Response 2: Thank you for your advice. We revised the manuscript according to the suggestion.

Point 3: Section 1. Introduction: My suggestion is to include a paragraph or two which must be dedicated to the Regulation concerning the International Carriage of Dangerous Goods by Rail (RID). Please also include the respective references (e.g., https://otif.org/en/?page_id=1105). I consider that it is very important for the reader to see the definitions and terminology concerning the topic “Dangerous Goods by Rail”.

Response 3: Thank you for your suggestion. We added two paragraphs to introduce the definitions and terminology concerning the topic “Dangerous Goods by Rail”. Thank you again for your  valuable suggestions to improve the quality of our manuscript. 

Point 4: Section 2. The Reliability Based Rescue Resource Scheduling Model: Please include a Data Flow Chart describing all your methodological steps so that the reader can obtain a clear overview of your work from the early beginning of the manuscript.

Response 4: Thank you for your advice. We added a research flowchart of in the Sections 2.

Point 5: Page 2, lines 76-78: There are 16 references which are associated with the specific sentence. My suggestion is to try to enhance the text which is associated with these references since their topics are very interesting.

Response 5: Thank you for your advice. We modify the description of this part.

Point 6: Page 3, lines 102 and 103, “Therefore, in case of railway dangerous goods accident, it is urgent to transport all kinds of rescue equipment and resources to the accident place.”. My suggestion is to provide analytical details concerning “rescue equipment and resources”. In addition, please justify why “all kinds…” are needed. Is it possible to having prioritization when considering the “rescue equipment and resources” taking into account the available technology for incident management?

Response 6: Thank you for your suggestion. We provided some rescue equipment and materials. When considering "rescue equipment and resources", it really needs to be prioritized. This paper only considers the emergency rescue materials when the accident occurs, and the subsequent materials for recovery and cleaning are not considered. Your suggestion points out a future research direction for us, thank you.

Point 7: Page 6, line 213: Please consider the idea to number the equations presented in this specific part of the manuscript (just above Section 3. Solution Algorithms).

Response 7: Thank you for your reminding. We numbered the equations in this part.

Point 8: Page 7, line 249: It seems that “Definition” has a different font height (points).

Response 8: Thank you for your reminding.

Point 9: Page 11, Section 4. Experimental analysis: Your experiment considered a regional road network. What would be the differences in your approach in the case of a semi urban road network where the basic traffic characteristics (traffic volume, speed, density) are more difficult?

Response 9: In the case of semi urban road network, road intersections generally have no signal control and there is less traffic flow in the road section. The basic traffic characteristics are difficult to obtain. However, in this case, the delay of intersection and road section is less, and these data may be obtained through emergency drill. So, our approach can be applied to the semi urban road network.

Point 10: Page 11, Table 5. The maximum tolerable transport time and the assembly time of each rescue center.: Please insert “n.a.” (not applicable or not available) in the empty cells of the specific Table.

Response 10: Thank you for your reminding, again.

Point 11: Please change Section “3 Solution Algorithms” to Section “3. Solution Algorithms”. Please change Section “4 Experimental analysis” to Section “4. Experimental Analysis”. Please change Section “5 Conclusions” to Section “5. Conclusions”.

Response 11: Thank you for your reminding.

Point 12: Section 5. Conclusions: Please include a summary with the constraints and limitations of your research. In addition, my suggestion is to formulate some policy recommendations arising from your findings and then to address these recommendations to the respective stakeholders (e.g., railway operators, rescue centers etc.). Finally, please include the future directions of the research on the topic of the article.

Response 12: Thank you for your suggestion. In the conclusions, we put forward some suggestions on emergency plan, personnel training and emergency drill from the perspective of emergency response time, material reserve and emergency path planning of emergency units. Moreover, the future directions of the research is put forward from two aspects.

Round 2

Reviewer 1 Report

The authors addressed the reviewers' comments and the paper is improved. I propose to publish the paper.

Reviewer 2 Report

The authors have successfully solved all my previous comments and concerns.

Reviewer 3 Report

I would like to express my deepest thanks to the authors because they have carefully addressed all my comments.